# High-pH structure of EmrE reveals the mechanism of proton-coupled substrate transport

Alexander A. Shcherbakov [1], Peyton J. Spreacker [2], Aurelio J. Dregni [1], Katherine A. Henzler-Wildman [2] & Mei Hong [1✉]

The homo-dimeric bacterial membrane protein EmrE effluxes polyaromatic cationic substrates in a proton-coupled manner to cause multidrug resistance. We recently determined the structure of substrate-bound EmrE in phospholipid bilayers by measuring hundreds of protein-ligand $H^N$–F distances for a fluorinated substrate, 4-fluoro-tetraphenylphosphonium ($F_4$-TPP$^+$), using solid-state NMR. This structure was solved at low pH where one of the two proton-binding Glu14 residues is protonated. Here, to understand how substrate transport depends on pH, we determine the structure of the EmrE-TPP complex at high pH, where both Glu14 residues are deprotonated. The high-pH complex exhibits an elongated and hydrated binding pocket in which the substrate is similarly exposed to the two sides of the membrane. In contrast, the low-pH complex asymmetrically exposes the substrate to one side of the membrane. These pH-dependent EmrE conformations provide detailed insights into the alternating-access model, and suggest that the high-pH conformation may facilitate proton binding in the presence of the substrate, thus accelerating the conformational change of EmrE to export the substrate.

[1] Department of Chemistry, Massachusetts Institute of Technology, 170 Albany Street, Cambridge, MA 02139, USA. [2] Department of Biochemistry, University of Wisconsin at Madison, Madison, WI 53706, USA. ✉email: meihong@mit.edu

Multidrug resistance (MDR) is a major public health concern that can undermine years of drug development efforts and result in epidemics of drug-resistant infections[1]. One of the mechanisms by which cells can become resistant to therapeutics is via expression of transmembrane (TM) efflux pump proteins in the small multidrug resistance (SMR) family. These SMR transporters remove cytotoxins from the cytoplasm by coupling the uphill efflux process to the downhill influx of protons across the cytoplasmic membrane (Fig. 1a)[2]. Unlike selective active transport proteins that recognize and transport a single substrate, the SMR proteins efflux a variety of cytotoxic compounds with different shapes, sizes, and chemical properties[3,4]. The relatively small size of these transporters was originally thought to provide a minimal model system for studying secondary active transport[5]. However, due to the sensitivity of these proteins to their environment, their conformational plasticity, and lack of extracellular domains, high-resolution structural information had been limited for many years[6–10]. The best studied SMR transporter, EmrE, is involved in pH and osmotic stress responses, biofilm formation, and resistance to a variety of poly-aromatic cations[11–14]. Biochemical and biophysical data have shown that the transport process of EmrE is highly complex. For example, in addition to acting as a proton-coupled antiporter, EmrE can also function as a proton-coupled symporter or uncoupled uniporter under different conditions[15–19]. Elucidating the mechanism of membrane transport by EmrE requires atomic-resolution structural information for multiple states of the protein, as well as dynamics information about the protein and the ligands throughout the transport cycle.

The 110-residue EmrE forms an antiparallel, asymmetric homo-dimer. Early electron microscopy (EM) data established that EmrE did not possess two-fold symmetry[9,20]. A subsequent 3.8 Å crystal structure showed that the two subunits are oriented in an antiparallel fashion, indicating that the protein has dual membrane topology[8]. This antiparallel conformation was later confirmed by NMR and single-molecule FRET data[21], mutagenesis[22–24], EPR data[25,26], and studies of homologous proteins[10,27]. Solution and solid-state NMR (ssNMR) data showed that the two subunits of the dimer are con-formationally asymmetric, exhibiting two sets of chemical shifts, and the TM helices undergo major reorientations as they exchange between the inward- and outward-facing states[6,16,21,28,29]. Cross-linking the antiparallel dimer blocked alternating acces in vitro and ethidium resistance in vivo, demonstrating the functional importance of this structural transition[29]. The proton-binding residue of EmrE is E14, which exhibits $pK_a$ values of $7.2 \pm 0.1$ and $8.4 \pm 0.2$ in the A and B subunits of the dimer in both lipid bilayers and bicelles[15,30]. Solution NMR experiments showed that the substrate, tetra-phenylphosphonium (TPP[+]), binds the protein asymmetrically, interacting primarily with one subunit and protecting that E14 from protonation, while E14 in the other subunit remains accessible to protonation with a $pK_a$ of 6.8[15,17]. This proton-binding asymmetry

was confirmed by magic-angle-spinning (MAS) NMR data that show that the E14 sidechain carboxyl chemical shifts differ between the two subunits[31]. A 2.3 Å crystal structure of the EmrE homolog, Gdx, was determined in complex with monobodies[10]. This structure confirms the asymmetry of the antiparallel dimer structure and the asymmetric interaction of the substrate with the key glutamate residues in the pore. However, Gdx is a selective guanidinium efflux pump, not a multidrug transporter-like EmrE, and biophysical data is more lim-ited for the Gdx transporter. To fully understand the molecular mechanism of EmrE transport, high-resolution structures are essential. This experimental structure information will enable an integrated analysis of the wealth of data on EmrE dynamics and function, allow assessment of the validity of molecular dynamics simulations[32,33] performed using the backbone-only EmrE crystal structure, and give insight into the similarities and differences between selective and non-selective transporters in the SMR family.

Recently, we discovered a single-point mutant, S64V-EmrE, that retains wild-type substrate-binding affinity but has slower rates of alternating access[7]. At the same time, we developed a $^1H–^{19}F$ REDOR NMR technique to measure distances to the ~2 nm range for structure determination[34–38]. Due to $^1H$ detec-tion under fast MAS, this technique has high spectral sensitivity, thus increasing the throughput of the distance measurement. Exploiting these biochemical and spectroscopic advances, we determined an experimental structure of S64V-EmrE complexed to a fluorinated substrate, $F_4$-TPP[+] (Fig. 1b) in DMPC bilayers at pH 5.8[39]. At this acidic pH, one of the two E14 residues is pro-tonated and neutral while the other residue remains deprotonated and anionic. By measuring ~200 protein–ligand distances as well as site-specific protein chemical shifts, we determined the struc-ture of this acidic-pH complex (abbreviated as EmrE-TPP below) to an average pairwise backbone root-mean-square deviation (RMSD) of 1.6 Å. The structure was calculated by docking the ligand to the protein, followed by molecular dynamics simula-tions that equilibrate the protein in explicitly solvated lipid bilayers, all under experimentally measured distance and torsion angle constraints. This low-pH structure shows that the cationic substrate lies closer to E14A than E14B, consistent with the asymmetric $pK_a$'s. The binding pocket is lined with numerous aromatic residues, including W63, Y60, F44, and Y40. These aromatic sidechains interact with the four ligand phenylene rings to stabilize the substrate, while still leaving sufficient space for the substrate to reorient. While this low-pH structure of EmrE gives a glimpse of the protein–substrate binding geometry, it is not suf-ficient for revealing the transport mechanism, because alternating access requires the protein to adopt multiple conformations throughout the transport cycle. Experimental data on multiple structural states is required to understand how drug binding and proton binding drive the conformational changes needed to transport the substrates across the membrane.

Here we determine the high-pH structure of the EmrE-TPP complex using the $^1H–^{19}F$ REDOR NMR experiment. We mea-sure ~380 protein–ligand H[N]–F distances, which combine with chemical-shift derived torsion angles to enable the calculation of the high-pH structure. We also investigate millisecond-timescale motion of $F_4$-TPP[+] at the binding site using 2D $^{19}F–^{19}F$ exchange NMR. These structural and dynamical results provide informa-tion about how changes in the protonation state of the protein drives structural transitions that enable EmrE to transport sub-strates in a proton-coupled manner.

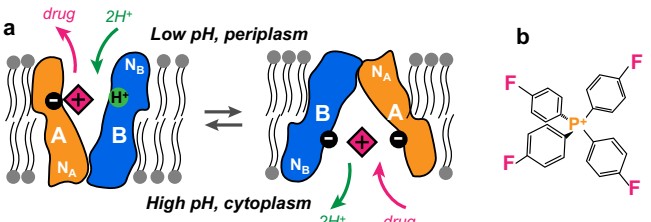

**Fig. 1 Schematic of the EmrE transport function. a** Simplified mechanistic model of pH-dependent substrate transport by EmrE. The binding-site structure of the cationic substrate in the dimeric protein depends on the protonation state of the proton-binding residue E14. **b** Structure of the substrate $F_4$-TPP[+].

## Results

**Resonance assignment of the high-pH EmrE-TPP complex.** To understand how proton binding and release change the structure of substrate-bound EmrE, we first ascertained the E14 $pK_a$ of

S64V-EmrE in complex with $F_4$-$TPP^+$. We previously carried out extensive pH titrations of $TPP^+$-bound EmrE in bicelles using solution NMR and found that the WT transporter had a single $pK_a$ of $6.8 \pm 0.1$[17], while S64V-EmrE had a single $pK_a$ of $7.0 \pm 0.1$[7]. We verified that this $pK_a$ was not significantly different when S64V-EmrE is bound to $F_4$-$TPP^+$ by repeating this pH titration. Global analysis of multiple residues yields a $pK_a$ value of $6.9 \pm 0.1$ (Supplementary Fig. 1). We therefore prepared a sample of S64V-EmrE bound to $F_4$-$TPP^+$ in DMPC bilayers at pH 8.0 to determine the structure of the unprotonated complex, to compare with the protonated complex previously determined at pH 5.8 in DMPC bilayers.

We first assessed the conformational homogeneity of the high-pH EmrE-TPP complex using 2D $^1H$–$^{15}N$ correlation (hNH) spectra (Fig. 2a). The spectra show a typical $^{15}N$ linewidth of ~0.5 ppm, which is narrower than the low-pH sample[39], indicating that the protein is structurally more homogeneous at high pH than at low pH. The $^{15}N$ and $^1H$ chemical shifts at high pH differ moderately from the low-pH values, which preclude the transfer of the chemical shifts from the low-pH spectra. Thus, we measured eight 3D $^1H$-detected correlation spectra (Fig. 2b, c and Supplementary Table 1) to independently assign the resonances of the high-pH complex. Among these experiments, the 3D hNcacoNH and HncacoHN spectra are particularly useful for sequence-specific assignment. Based on the peak connectivities, we obtained four backbone chemical shifts (Cα, CO, $N^H$, $^HN$) for 77 residues in subunit A and 64 residues in subunit B. In addition, 64 residues in subunit A and 53 residues in subunit B have assigned Cβ chemical shifts. The resulting secondary chemical shifts confirm the presence of four TM helices in the protein (Supplementary Fig. 2). However, the high-pH EmrE exhibits significant chemical shift differences from the low-pH protein in the TM3 and TM1 helices of subunit B (Supplementary Fig. 3). These two TM helices contain the important functional residues W63 and E14, respectively, suggesting that the protein interacts with the substrate differently between high and low pH. Between subunits A and B, TM1 and TM3 helices have larger chemical shift differences compared to TM2 and TM4 helices, similarly indicating the importance of TM1 and TM3 helices for ligand binding (Supplementary Fig. 4).

**Measurement of protein-ligand $H^N$–F distances**. The assignment of the $H^N$ and $^{15}N$ chemical shifts allowed us to measure and resolve protein-substrate $H^N$–F distances using the $^1H$-detected and hNH-resolved $^1H$–$^{19}F$ REDOR experiment[37]. We measured the 2D REDOR spectra at three mixing times, 1.68, 2.53, and 3.78 ms. For each mixing time, a control 2D $S_0$ spectrum with $^{19}F$ pulses off and a dephased $S$ spectrum with $^{19}F$ pulses on were measured. The former exhibits all backbone and sidechain $H^N$ signals (Fig. 3a) whereas the latter shows weaker intensities for amide protons that experience significant $^1H$–$^{19}F$ dipolar couplings. The difference spectrum ($\Delta S$) between $S_0$ and $S$ thus manifests only the signals of amide protons that are near the fluorines. At 1.68 ms, we observed signals only from the nearest residues to $TPP^+$, such as E14, Y40, Y60, and W63, whereas at longer mixing times, signals from more remote residues such as G9, G57, and S75 are also detected. No difference signals are observed for residues C-terminal to the TM3 helix and for loop residues.

Fitting the intensity ratios $S/S_0$ between the REDOR $S_0$ and $S$ spectra allowed us to extract precise $H^N$–F distances (Fig. 3b and Supplementary Fig. 5). We observed significant dipolar dephasing for many residues. Among 116 resolved dipolar dephasing curves, 36 show REDOR dephasing that corresponds to distances of less

than 9 Å. The shortest distance is found for W63B indole Nε, which is 3.8 Å from the nearest fluorine. In general, TM2 and TM3 residues have some of the closest contact with $F_4$-$TPP^+$. For example, the S43A $H^N$ is 4.3 Å from the nearest $^{19}F$. The distances in the high-pH complex differ substantially from the low-pH values. For example, A10B, S43B, and W63Bε are closer to the ligand fluorines by 1.2–2.5 Å while T18A and W63A backbone amides are further from the ligand fluorines by 0.2–0.3 Å.

**Ligand docking and structure calculation**. Using rigid-body docking (Supplementary Fig. 6a), we disambiguated the four-fold degeneracy of the fluorines and assigned the measured dipolar couplings to specific protein amide protons (Supplementary Tables 2 and 3). For weak REDOR dephasing that corresponds to distances longer than 10 Å, four constraints were created where the protein $H^N$ atom must be at least 10 Å away from each of the four fluorines. In total, we obtained 387 protein–ligand distance restraints from this docking analysis (Supplementary Table 4). $F_4$-$TPP^+$ docks to a single location between the two subunits, analogous to the low-pH complex. But the ligand orientation relative to the protein differs substantially from the low-pH structure. At high pH, three of the four phenylene ring planes are approximately parallel to the bilayer normal. In comparison, at low pH, only one phenylene ring is tangential to the bilayer normal whereas three rings lie transverse to the bilayer normal (Fig. 4a, b). Using two lowest-violation docked structures, we refined the protein structure using molecular dynamics simulations under the protein-ligand distance constraints, 148 pairs of (ϕ, ψ) angles and 76 $\chi_1$ torsion angles (Table 1 and Supplementary Table 5). The simulations equilibrated by 200 ns to a backbone RMSD of $2.85 \pm 0.95$ Å for the protein (Supplementary Fig. 6b) and $1.30 \pm 0.64$ Å for the $F_4$-$TPP^+$ phosphorous and its four directly bonded carbons (Table 1). Among the four TM helices, TM4 is the furthest away from the ligand: all resolved dipolar couplings correspond to distances of longer than 9 Å (Supplementary Fig. 5). This is consistent with the low-resolution crystal structure, which shows that TM1–TM3 form the substrate-binding pocket whereas the TM4 helices that control dimerization are away from the transport pore[8]. Since TM4 is not well constrained by the measured distances (Supplementary Fig. 6c), when we consider only the TM1–3 helices, the calculated structure has an improved backbone RMSD of $1.97 \pm 0.67$ Å.

**Structural differences between the high-pH and low-pH complex**. Interestingly, the high-pH EmrE-TPP complex exhibits numerous structural differences from the low-pH complex (Fig. 4 and Supplementary Fig. 7). First, the ligand is more symmetrically positioned between the two E14 residues at high pH: the distances between the $F_4$-$TPP^+$ phosphorus (P) and the two E14 Cδ carbons are $6.6 \pm 0.7$ Å to subunit A and $7.5 \pm 0.5$ Å to subunit B. In comparison, at low pH the P atom is 1.9 Å closer to the negatively charged E14A than to the neutral E14B (Table 2 and Fig. 4e, f). Thus, the protonation states of the two E14 residues directly impact the substrate position. Second, the inter-subunit proximities and relative orientations of the TM helices have changed between the high-pH and low-pH complexes. At high pH, the two E14-bearing TM1 helices are further away from each other, while the two W63-bearing TM3 helices are more parallel to each other (Fig. 4a, b and Supplementary Fig. 7) compared to the low-pH complex. The E14A CA–E14B CA distance is $16.9 \pm 0.6$ Å at high pH and shortens to $15.7 \pm 0.8$ Å at low pH. Overall, the TM helices become more parallel to each other at high pH, creating an elongated binding cavity in which $F_4$-$TPP^+$ is oriented with three

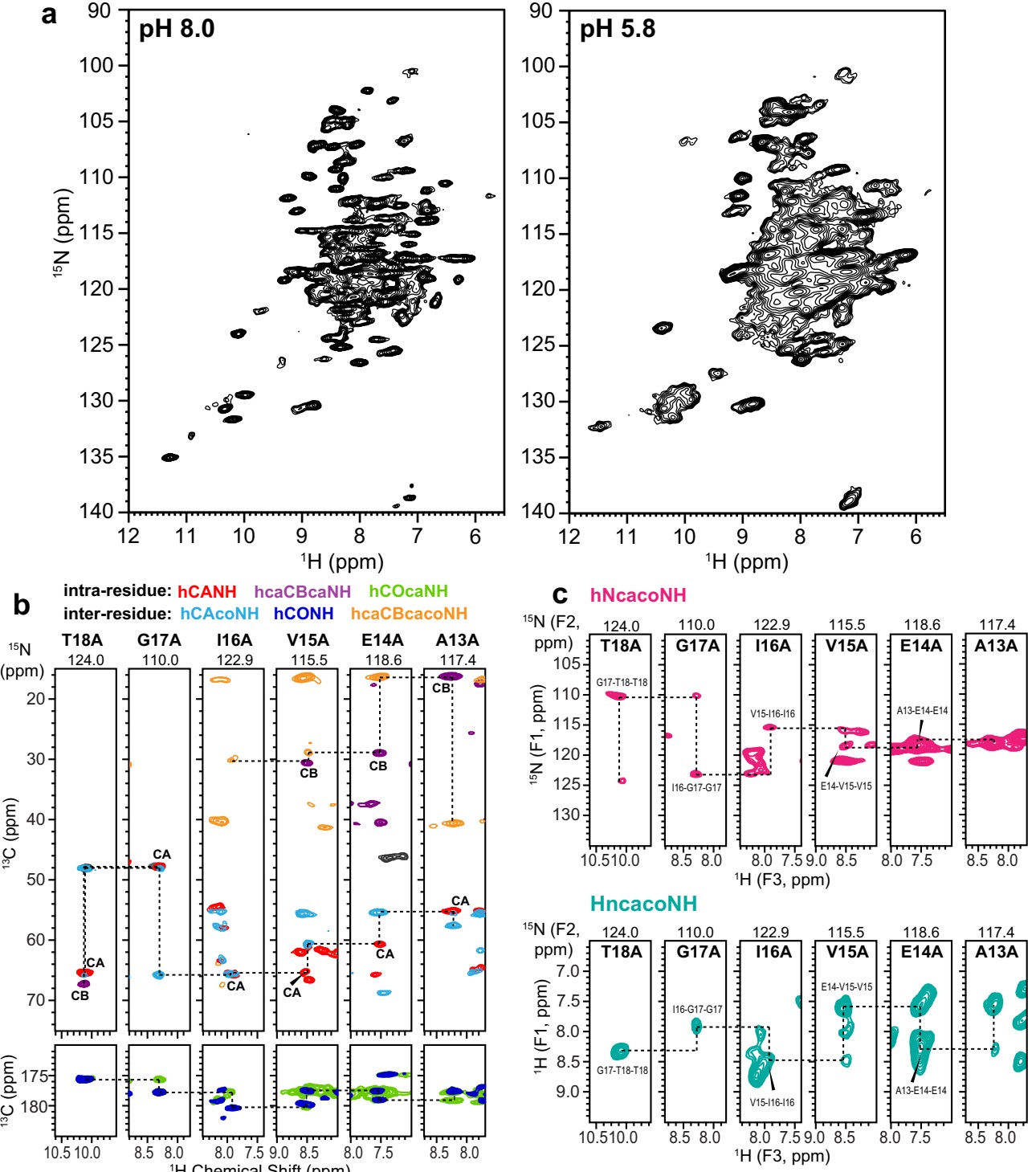

**Fig. 2 2D and 3D ¹H-detected MAS correlation spectra of TPP-bound EmrE in DMPC bilayers for resonance assignment. a** 2D hNH spectra of the protein at pH 8.0 compared to pH 5.8. The spectral linewidths are narrower at high pH, indicating higher structural homogeneity. **b** Six ¹H–¹³C–¹⁵N 3D correlation spectra to obtain intra-residue and inter-residue peak connectivities. The strips for residues T18 to A13 in subunit A are shown. The aliphatic ¹³C strips overlay four 3D spectra whereas the carbonyl ¹³C strips overlay two 3D spectra. Negative intensities in the hcaCBcaNH and hcaCBcacoNH spectra are shown in gray. Glycine residues show a negative CA peak rather than a positive CB peak. **c** Inter-residue 3D NNH and HNH correlation spectra. All spectra were measured under 55 kHz MAS on a 600 MHz NMR spectrometer.

out of four phenylene rings tangential to the bilayer normal. In comparison, at low pH the TM helices are oriented at very different angles with respect to each other. In particular, the N-terminal end of TM3A approaches the C-terminal end of TM3B (Supplementary Fig. 7b), which closes off water access on one side of the helical bundle. This closed-on-one-side configuration pushes the ligand towards the opposite end of the helical bundle, where the C-terminal end of TM3A is now splayed open from the N-terminal end of TM3B. The resulting shallow and open binding site at low pH exposes the ligand (Supplementary

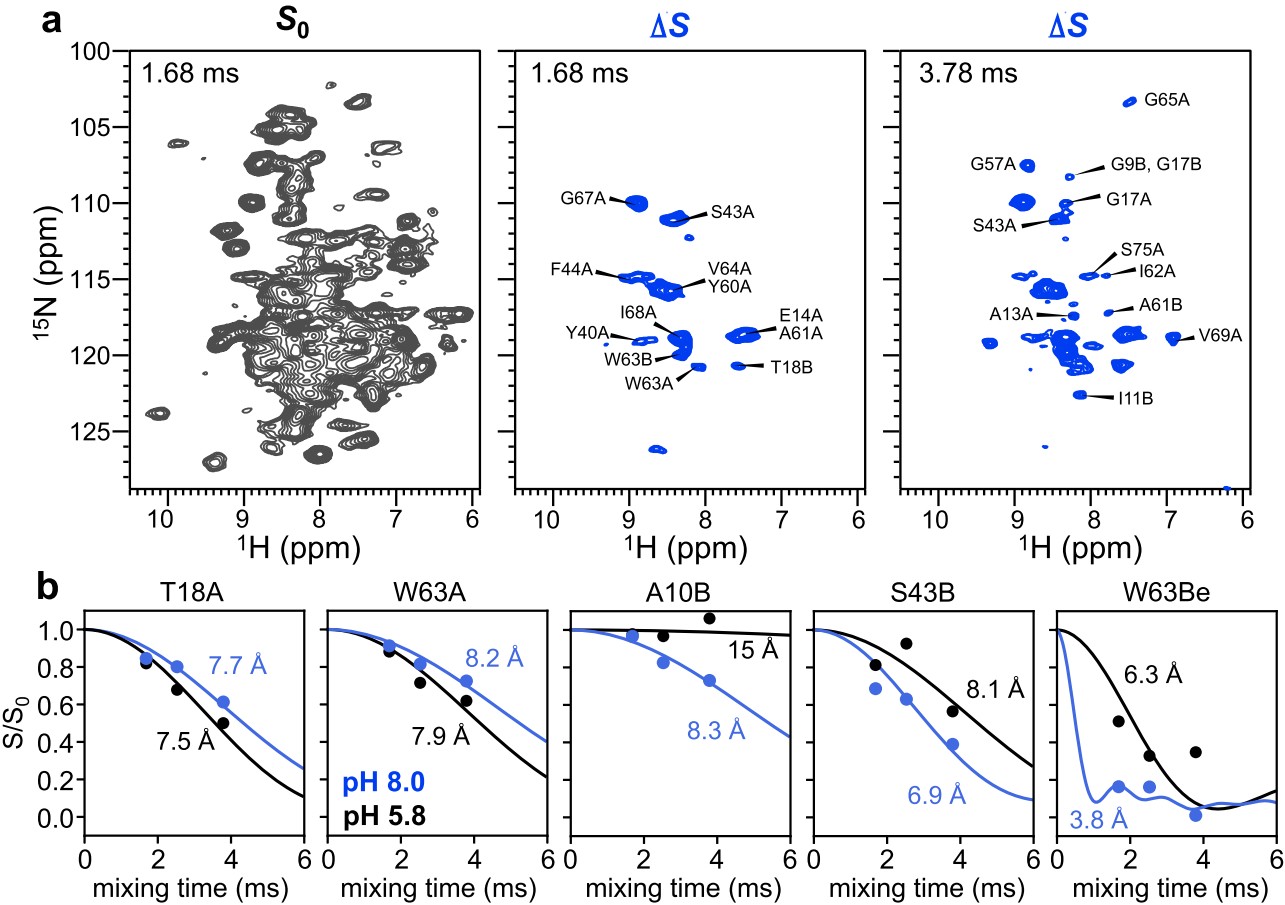

**Fig. 3 ¹H–¹⁹F distance measurements between EmrE H$^N$ and F$_4$-TPP$^+$ using 2D-hNH resolved ¹H–¹⁹F REDOR. a** Representative $S_0$ (black) and $\Delta S$ (blue) spectra for two mixing times, 1.68 and 3.78 ms. Assignment is shown for peaks in the $\Delta S$ spectrum. As the mixing time increases, more difference peaks are observed, corresponding to H$^N$ sites that are further from the fluorinated substrate. **b** Representative ¹H–¹⁹F REDOR $S/S_0$ dephasing curves with best-fit simulations. ¹H–¹⁹F distances at pH 8.0 (blue) differ from the pH 5.8 data (black). The indole H$^N$ of W63B has much shorter distances to the substrate fluorines at pH 8.0 than at pH 5.8. REDOR dephasing values are provided as a Source Data file.

Fig. 7b), allowing three of the four phenylene rings to lie transverse to the bilayer normal.

Compared to the low-pH complex, the high-pH EmrE-TPP complex has a longer and more symmetric binding cavity that is not fully closed on either side. Because the binding pocket is populated by protein sidechains, we next investigated which binding site is more spacious by measuring substrate dynamics using ¹⁹F NMR. This is a more direct and functional probe of the binding-site volume compared to computing the volume based on the structure. One-dimensional ¹⁹F direct-polarization (DP) NMR spectra of the substrate at high pH resolve three peaks with intensity ratios of 1:1:2 (Fig. 5a), indicating that the four fluorines of the ligand reside in distinct chemical and conformational environments. The most downfield peak (peak 4) has a narrow linewidth of 0.8 ppm (~450 Hz), indicating that one of the fluorines resides in a well-ordered structural environment. The presence of one narrow ¹⁹F peak is also observed at low pH, but this narrow peak has the most upfield chemical shift. In both cases, ¹H–¹⁹F cross-polarization (CP) spectra preferentially enhanced the intensity of the narrow peak relative to the other ¹⁹F signals, indicating that this fluorine is the most immobilized. Compared to the low-pH sample, the high-pH ¹⁹F chemical shifts of F$_4$-TPP$^+$ shifted by about 6 ppm downfield, indicating that the binding-site aromatic residues interact with the substrate very differently at high pH.

**Dynamics and hydration of the ligand at the binding site.** To probe millisecond-timescale motions of the ligand in the binding pocket, we measured 2D ¹⁹F–¹⁹F exchange spectra of F$_4$-TPP$^+$ at 285 K using mixing times of 0.1 ms to 80 ms (Fig. 5b). Exchange cross peaks are readily observed by ~20 ms at this temperature, but are absent at 265 K[39], indicating that exchange on this timescale is due to motion rather than ¹⁹F spin diffusion. The diagonal intensity decays and cross-peak intensity buildup occur with rates of 165–318 s$^{-1}$ (Fig. 5c and Supplementary Fig. 8). These rates are about two-fold faster than the low-pH rates (75–103 s$^{-1}$), indicating that the ligand is more dynamic in the high-pH complex. These increased dynamics at high pH agree well with the measured thermodynamic parameters for ligand binding. EmrE is a proton-coupled transporter, and we have previously shown that protons are released from both E14 and H110 upon TPP$^+$-binding[18]. ITC experiments were performed using multiple buffers with different ionization enthalpies to determine the number of protons released. These experiments were performed at four different pH values. This data was analyzed to extract the enthalpy and entropy of binding independent of the buffer contribution by extrapolating to $\Delta H_{ionization} = 0$ (Supplementary Fig. 9). The resulting thermodynamic parameters are shown in Table 3. This data shows that the well-established increase in binding affinity at high pH is driven by an increasingly favorable entropic contribution, while the enthalpy of binding becomes less favorable.

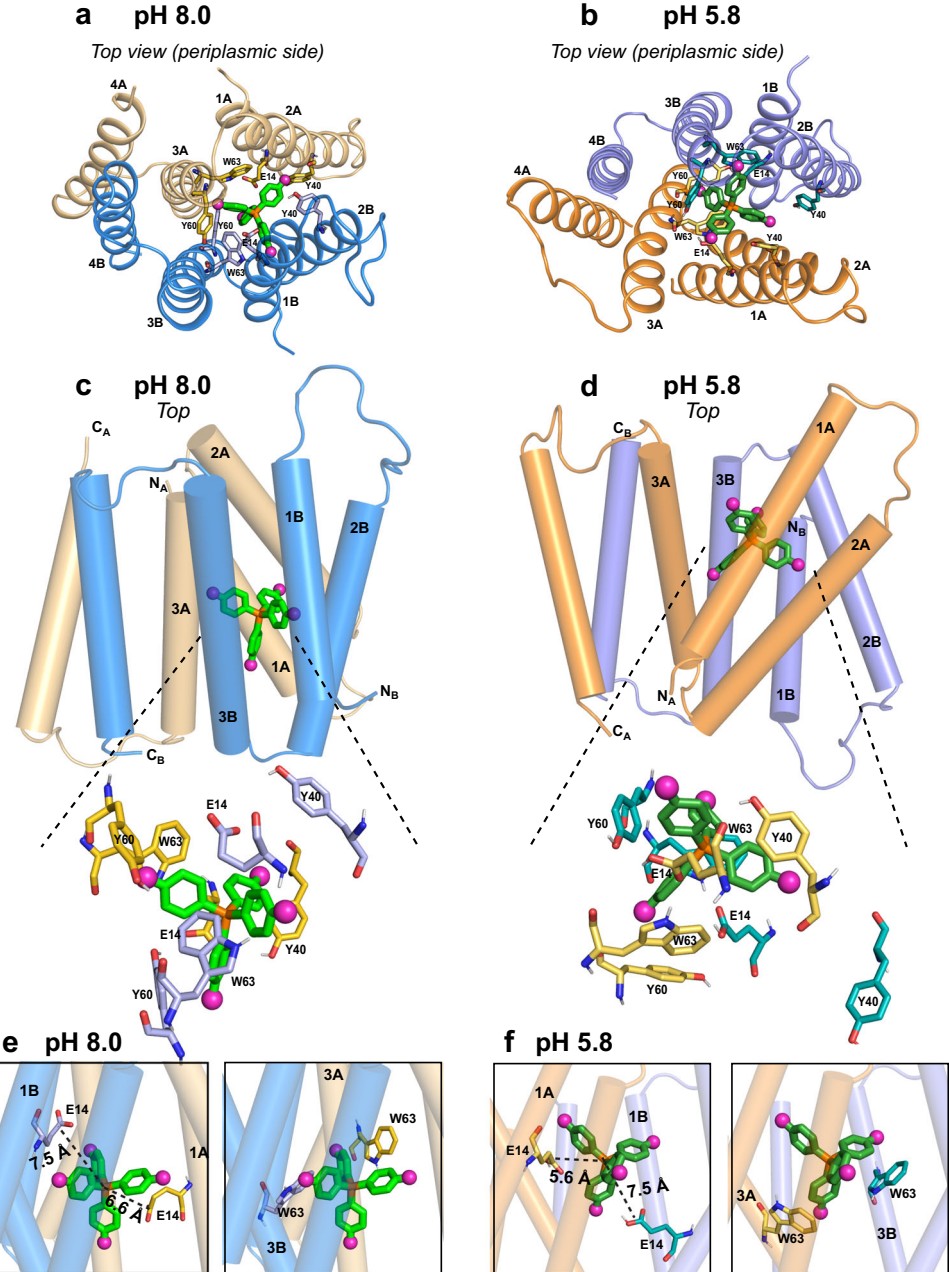

**Fig. 4 Structure of the EmrE-TPP complex in DMPC bilayers at high pH (PDB: 7SFQ) and its comparison with the previously determined low-pH structure (PDB: 7JK8). a** Top view of the high-pH EmrE-TPP complex structure, seen from the periplasmic side. The crucial binding-site residues E14, Y40, Y60, and W63 are shown as sticks. **b** Top view of the low-pH EmrE-TPP complex for comparison. Note that conformer A (beige) in the high-pH complex has changed to conformer B (purple) in the low-pH complex. This conformational change switches the designation of the two subunits between the high and low pH complexes. **c** Side view of the high-pH structure. **d** Side view of the low-pH structure. The ligand is buried deep in the high-pH complex but is more exposed to the top side in the low-pH complex. The arrangement of the TM helices differs noticeably between the two structures. **e** TPP⁺ position relative to E14 and W63 in the high-pH structure. The ligand center P atom is similarly distanced from the two E14 residues. **f** TPP position relative to E14 and W63 in the low-pH structure. The ligand is ~2 Å closer to E14A than E14B. .

The faster ligand reorientations in the high-pH complex suggest that the binding pocket should be more hydrated compared to the low-pH complex. To test this hypothesis, we measured water-edited 2D hNH spectra of the protein using water-to-protein $^1$H polarization transfer times of 30 and 325 ms (Fig. 6). The short-mixing-time spectra selectively exhibit well-hydrated residues. After correcting for $^1$H $R_1$ relaxation (Supplementary Fig. 10), we find that the high-pH complex exhibits higher water-transferred intensities than the low-pH protein. In particular, the TM1B and TM3B helices in subunit B

are much more water accessible at high pH than at low pH. Moreover, the extent of hydration is more comparable between the two subunits at high pH compared to the low-pH complex. This increased similarity of the hydration extent at high pH is exemplified by the G26 pair, and is consistent with the smaller chemical shift asymmetry of the protein at high pH. We attribute these observations to the more parallel orientations of the TM helices in the high-pH complex, which reduce the difference in the degree of opening between the two ends of the helical bundle. In comparison, the low-pH EmrE-TPP complex shows much

**Table 1 NMR and refinement statistics for $F_4$-TPP$^+$ bound S64V-EmrE structure in DMPC bilayers at pH 8.0.**

|  | Monomer A | Monomer B |
|---|---|---|
| *NMR distance and dihedral constraints* |  |  |
| Dipolar coupling measurements | 63 | 52 |
| Distance constraints | 205 | 182 |
| Total dihedral-angle restraints |  |  |
| $\phi$ | 83 | 65 |
| $\psi$ | 83 | 65 |
| $\chi_1$ | 43 | 33 |
| *Structure statistics* |  |  |
| Violations (mean ± s.d.) |  |  |
| Distance constraints (Å) | 0.004 ± 0.042 | 0.017 ± 0.112 |
| Max. distance-constraint violation (Å) | 0.85 | 1.62 |
| $\phi$ Dihedral-angle constraints (°) | 0.302 ± 1.815 | 0.545 ± 3.454 |
| $\psi$ Dihedral-angle constraints (°) | 0.744 ± 4.289 | 0.508 ± 3.905 |
| Max. $\phi$ dihedral-angle violation (°) | 17.2 | 31.9 |
| Max. $\psi$ dihedral-angle violation (°) | 53.8 | 60.1 |
| *Average pairwise r.m.s.d (Å)[a]* |  |  |
| Protein heavy atom | 3.43 ± 1.06 |  |
| Protein backbone | 2.85 ± 0.95 |  |
| Protein transmembrane heavy atom | 2.66 ± 0.86 |  |
| Protein transmembrane backbone | 2.23 ± 0.83 |  |
| Ligand heavy | 1.84 ± 0.61 |  |
| Ligand center[b] | 1.30 ± 0.64 |  |

[a]Pairwise RMSD was calculated among 10 lowest-violation structures between the two independent MD runs after the refinement had equilibrated.
[b]Ligand center is operationally defined as phosphorus and the four directly bonded carbon atoms of $F_4$-TPP$^+$.

**Table 2 Protein-substrate distances extracted from the NMR-refined structural models.**

|  | pH 8.0 | | pH 5.8 | |
|---|---|---|---|---|
|  | Monomer A | Monomer B | Monomer A | Monomer B |
| P–E14 Cδ | 6.6 ± 0.7 Å | 7.5 ± 0.5 Å | 5.6 ± 0.3 Å | 7.5 ± 1.0 Å |
| P–Y40 Oζ | 7.6 ± 1.8 Å | 11.2 ± 1.2 Å | 6.8 ± 0.5 Å | 16.7 ± 0.3 Å |
| P–Y60 Oζ | 9.0 ± 1.2 Å | 6.9 ± 1.1 Å | 9.8 ± 0.7 Å | 5.9 ± 0.4 Å |
| P–W63 Nε | 5.6 ± 0.5 Å | 5.7 ± 0.9 Å | 6.0 ± 0.4 Å | 5.6 ± 0.3 Å |
| Min. F[a]–E14 Cδ | 4.7 ± 0.8 Å | 5.2 ± 0.6 Å | 4.6 ± 0.5 Å | 6.5 ± 0.7 Å |
| Min. F[a]–Y40 Oζ | 6.0 ± 1.0 Å | 9.5 ± 0.8 Å | 6.2 ± 0.6 Å | 12.2 ± 0.4 Å |
| Min. F[a]–Y60 Oζ | 6.1 ± 0.8 Å | 5.7 ± 0.6 Å | 6.9 ± 0.4 Å | 5.6 ± 0.5 Å |
| Min. F[a]–W63 Nε | 4.8 ± 0.3 Å | 4.9 ± 0.7 Å | 5.8 ± 0.3 Å | 5.7 ± 0.4 Å |

The average distances and standard deviations are from the ensemble of 10 minimum constraint-violating structures in the final 240 ns of MD trial 1 and 160 ns of MD trial 2. P refers to TPP phosphorus atom while protein atom is denoted by standard IUPAC nomenclature.
[a]Distances of the nearest fluorine to specific protein atoms.

higher hydration for subunit A than subunit B, in good agreement with the larger conformational asymmetry of the two subunits. These spectral observations are borne out by the MD equilibrated structural ensemble, as the structure calculation explicitly solvated the protein-ligand complex in lipid bilayers. Figure 6e shows membrane-embedded water molecules whose oxygen atoms lie within 15 Å of any of the ligand atoms. Strikingly, the ligand-binding site is much more hydrated in the high-pH complex: a total of 69 water molecules are found in the ligand-binding pocket, and these water molecules approach the ligand from both sides of the lipid bilayer. In comparison, only 23 water molecules are found in the ligand-binding pocket in the low-pH complex; moreover, they approach the ligand only from one side, the putative periplasmic side, of the lipid bilayer.

## Discussion

The high-pH structure the EmrE-TPP complex is consistent with the available biochemical data and suggests a mechanism for how proton binding drives release of high-affinity substrates from this promiscuous transporter. Extensive mutagenesis of residues throughout EmrE have demonstrated that E14 in TM1 is critical for binding both drug-like substrates and protons for the transport activity[40–42]. Residues in TM2 and TM3 are important for substrate binding and substrate specificity[4,43–45]. TM3 also contains a putative hinge region that is important for controlling the rate of alternating access of the transporter. TM4 is outside the binding pocket and contains the dimerization motif that stabilizes this highly dynamic homodimer[24,32,46]. The measured substrate-protein distances shown here and in the previous study are the shortest for residues in TM1–3 at both low and high pH, consistent with the substrate-binding site inferred from mutagenesis and the overall structural organization of the transporter observed in low-resolution EM[9] and crystal structures[8]. The asymmetry of chemical shifts between subunits A and B is the largest in TM3 (Supplementary Fig. 4). This is consistent with the proposal that the hinge and the difference in local conformation of this helix[21] between the two subunits control the asymmetric structure of the EmrE dimer and the rate of alternating access.

The fact that these significant structural changes result exclusively from a pH change is remarkable. Deprotonation of the two

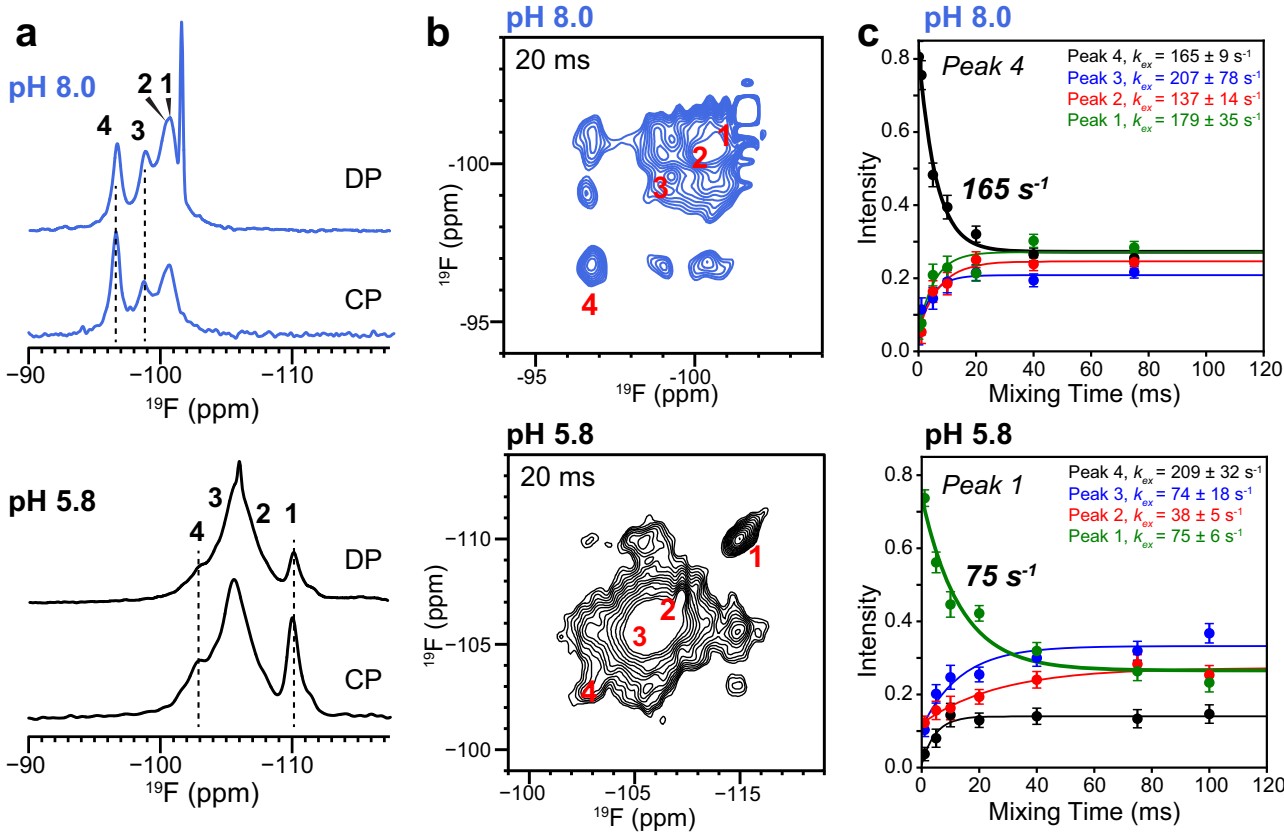

**Fig. 5 $^{19}$F NMR spectra of F$_4$-TPP$^+$ to probe substrate structure and dynamics. a** 1D $^{19}$F NMR spectra of F$_4$-TPP$^+$ bound to EmrE at pH 8.0 and pH 5.8. Multiple $^{19}$F chemical shifts are resolved, indicating heterogeneous structural environments of the four fluorines. **b** 2D $^{19}$F–$^{19}$F correlation spectra of EmrE-bound F$_4$-TPP$^+$ at pH 8.0 (blue) and pH 5.8 (black), measured with a mixing time of 20 ms. **c** $^{19}$F exchange buildup and decay curves for the resolved peak 4 at pH 8.0 and the resolved peak 1 at pH 5.8. Data are presented as mean values +/− 2σ. Error of intensity values was propagated from spectral signal to noise, while fitting parameter errors were estimated by Monte Carlo methods. These $^{19}$F spectra were measured under 38 kHz MAS at a sample temperature of ~285 K. Cross-peak intensities are provided as a Source Data file.

**Table 3 pH-dependent binding parameters determined by ITC.**

| pH | $\Delta G$ (kJ/mol) | $\Delta H$ (kJ/mol) | $-T\Delta S$ (kJ/mol) |
|---|---|---|---|
| 5.5 | −25.3 ± 0.3 | −40.0 ± 0.6 | 15 ± 0.7 |
| 6.5 | −34.9 ± 0.4 | −31.5 ± 1.7 | −3.4 ± 1.7 |
| 7.5 | −41.1 ± 0.8 | −19.1 ± 0.6 | −21.0 ± 1.0 |
| 8.5 | −46.0 ± 0.6 | −20.7 ± 2.7 | −25.3 ± 2.8 |

Original data for TPP$^+$ binding to WT EmrE was reported in[18]. Here we present the average pH-dependent binding affinity, enthalpy, and entropy for substrate–protein interaction, independent of any contribution from buffer ionization.

E14 residues not only symmetrizes the TPP$^+$ position but also increases the separation between the two TM1 helices. The E14 Cδ-Cδ distance increased from 11.8 ± 1.3 Å at low pH to 13.3 ± 0.8 Å at high pH. This subtle increase of the E14–E14 separation, and the ensuing change in the binding pocket geometry (Supplementary Fig. 7), translates to a noticeable effect on the drug dynamics, speeding up ligand reorientation rates two-fold compared to the low-pH complex. The motion is likely tetrahedral jumps. 2D $^{19}$F–$^{19}$F exchange data suggest that this tetrahedral jump is incomplete in geometry: while peak 4 displays equilibrated exchange intensities of ~0.25 by 80 ms, the other three sites do not fully equilibrate, suggesting that these fluorines are impeded by the protein sidechains. The substrate motion is more complete in the high-pH complex than in the low-pH complex, whose $^{19}$F exchange intensities are less

equilibrated (Supplementary Fig. 8d). The differences between the structures and drug dynamics of the low- and high-pH states indicate that ligand dynamics depend both on the size of the binding pocket and on sidechain obstructions.

Comparison of the structures of the EmrE-TPP complex at low and high pH is most informative in understanding how protonation of one E14 residue can drive release of the TPP$^+$ substrate. It is well established that the apparent affinity of EmrE for drug-like substrate is weaker at low pH than at high pH[42], and this is due to a faster substrate off-rate at low pH. This pH-dependent substrate affinity was originally attributed to a simple competition between TPP$^+$ and proton for binding to E14. However, we now know that EmrE can bind proton and TPP$^+$ simultaneously at low pH[17,39], and the two E14 residues are sufficiently far apart from each other that electrostatic coupling is minimal and proton binding and release occur relatively independently of either residue[30]. Examining the structural changes of the EmrE–TPP complex from high pH to low pH immediately suggests why TPP$^+$ affinity is lower in the protonated state. At high pH, TPP$^+$ is buried deep within the helical bundle, positioned nearly symmetrically between the two TM1 helices within an elongated binding cavity (Fig. 4 and Supplementary Fig. 7). In contrast, at low pH, the transporter structure is more asymmetric and clearly closed on one side and open on the other. TPP$^+$ is positioned much closer to the open end of the binding cavity, primed for release. The weakening of substrate affinity upon proton binding is likely important for speeding up the release of tight-binding substrates so that this promiscuous transporter can rapidly efflux

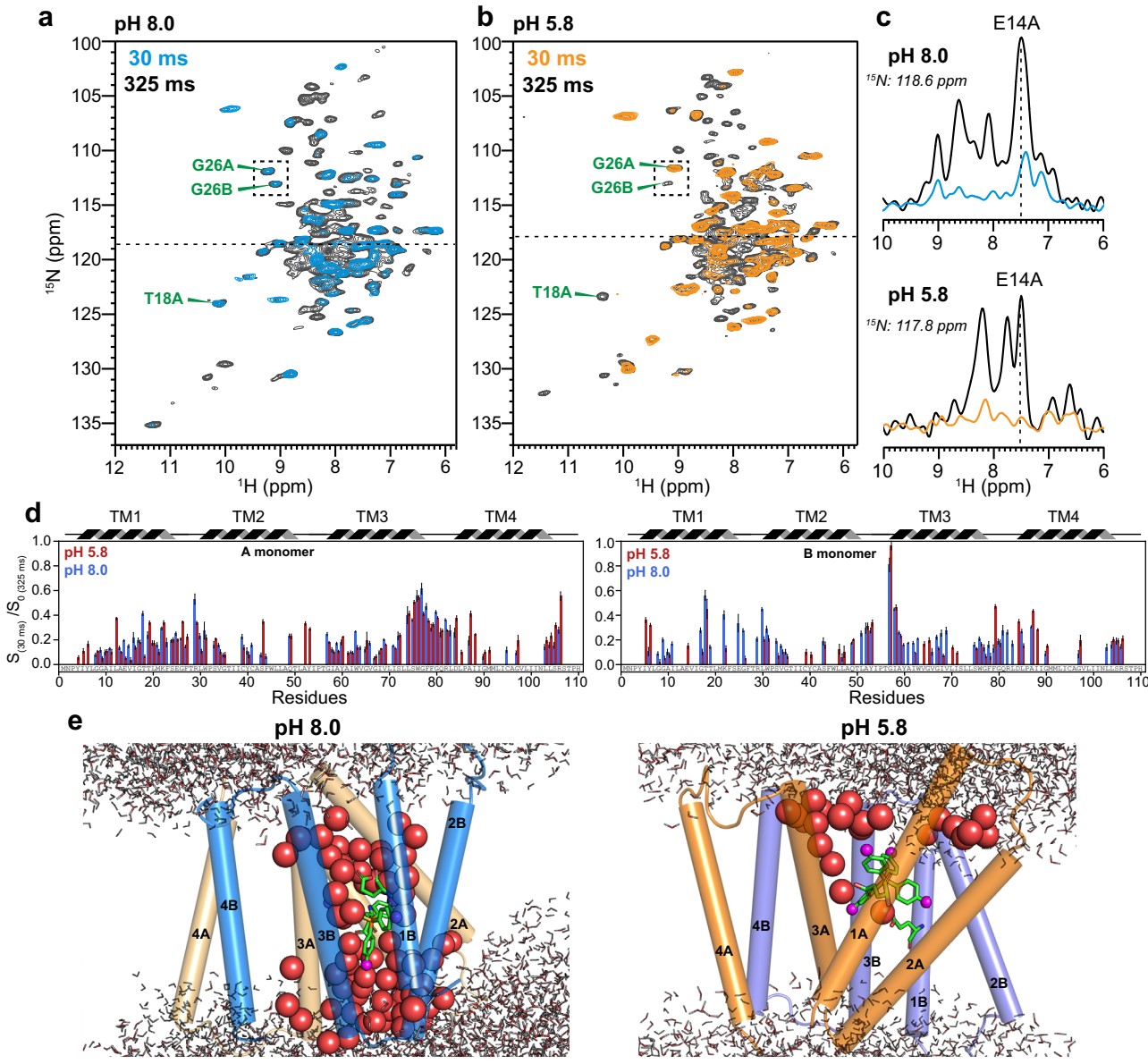

**Fig. 6 Water-edited 2D hNH spectra of TPP+-bound EmrE. a** pH 8.0 spectra. **b** pH 5.8 spectra. **c** 1D $^{15}$N cross-sections of E14A extracted from the water-edited 2D hNH spectra. The pH 8.0 sample shows higher water-transferred intensities than the pH 5.8 sample. **d** Intensity ratios $S/S_0$ between the 30 and 325 ms spectra of the water-edited 2D spectra to compare the water accessibilities between the high and low pH proteins. The intensities have been corrected for $^1$H $R_1$ relaxation. **e** MD equilibrated membrane-bound structures of the EmrE-TPP+ complex at pH 8.0 and pH 5.8. Water oxygens within 15 Å of any of the ligand atoms and that are within the lipid bilayer are shown as red spheres. Membrane-surface water molecules are shown as thin lines, and water molecules >15 Å away from the protein are shown in gray. The number of water molecules is 69 in the high-pH complex but only 23 in the low-pH complex, consistent with the higher water-transferred intensities of the protein at high pH. Moreover, water hydrates the ligand from both sides of the membrane in the high-pH complex, but only accesses the ligand from the periplasmic side in the low-pH complex. The water-edited intensities are provided as a Source Data file.

a broad range of substrates with widely varying affinities[17,19]. The pH-dependent thermodynamic parameters (Table 3) show that higher-affinity drug binding by this promiscuous transporter is driven by increasingly favorable binding entropy. This is consistent with the enhanced ligand dynamics and hydration observed here in the high-pH structure. There is not sufficient ITC data or in vitro transport data available to assess whether the relative promiscuity of EmrE changes appreciably with pH. That remains an open question for further study.

The native environment for EmrE is the inner membrane of *E. coli*. This is an asymmetric environment unlike the in vitro conditions used for structure determination. Usually, the

periplasmic pH is lower than the cytoplasmic pH[47]. Given the significant structural changes of the protein we determined here as a function of E14 protonation state in a symmetric membrane, it is reasonable to assume that the protonation state of E14 regulates the overall structural change of the protein. We can then infer what the structure might look like in the presence of a transmembrane pH gradient. Cytoplasmic pH in *E. coli* is generally 7.4–7.8[48], above the pK$_a$ of E14 in the F$_4$-TPP+-bound transporter. Thus, the open-in conformation of EmrE is expected to have predominantly deprotonated E14 residues and more closely resemble the high-pH structure (Fig. 4a, c, e). In contrast, the periplasmic pH closely correlates with the pH of the external

environment[47]. Thus, when *E. coli* are in an acidic environment like the human gut, the periplasmic pH will be well below the p$K_a$ of E14 and the structure of EmrE should more closely resemble the low pH structure (Fig. 4b, d, f).

It is interesting to note that the high-pH structure of the EmrE-TPP complex is not fully closed on either side of the membrane. This is borne out by the water-accessibility data (Fig. 6a–d) and the MD simulations (Fig. 6e), which show larger water accessibility of the ligand-binding cavity at high pH than at low pH. Moreover, water molecules approach the ligand from both sides of the membrane at high pH, but only access the ligand from one side of the membrane at low pH. The dual-open conformation at high pH suggests that after a toxic substrate has bound to the protein from the cytoplasm and EmrE adopts a conformation similar to Supplementary Fig. 7a, there is sufficient space for protons to enter and bind E14 from the periplasmic side. This could then trigger a conformational change to a state similar to the low-pH structure, with the transporter open to the periplasmic side of the membrane. In this outward-facing state, the substrate is bound peripherally to the central transport pore, and is thus primed for release in accord with the enhanced off rate in the drug- and proton-bound state.

## Methods

**S64V-EmrE expression and purification**. S64V-EmrE was expressed and purified as previously described[7] using the same procedure as for WT EmrE[49]. $^2$H,$^{13}$C,$^{15}$N (CDN)-labeled S64V-EmrE was expressed in $^2$H$_2$O media containing 2.5 g/L U–$^2$H,$^{13}$C glucose, 1 g/L $^{15}$NH$_4$Cl, and 0.5 g/L $^2$H,$^{13}$C,$^{15}$N-labled ISOGRO. Lysis and purification were performed as previously reported[50,51] using Ni-NTA affinity column followed by thrombin cleavage of the His-tag and size exclusion chromatography on a S200 column in buffer containing 50 mM MES, 20 mM NaCl, 10 mM decyl-maltoside, 5 mM BME, pH 7.0.

**Solution NMR spectroscopy and p$K_a$ analysis**. Solution NMR data were collected on samples with 1.0 mM $^2$H, $^{15}$N S64V-EmrE in DMPC/DHPC bicelles ($q = 0.33$, with a protein to DMPC molar ratio of 1:50). The buffer contained 20 mM NaCl, 50 mM sodium acetate, 50 mM MOPS, 50 mM MES, 50 mM bicine, 2 mM TCEP, 0.01% DSS, and 10% D$_2$O. About 10 mg F$_4$-TPP$^+$ was added to this protein bicelle solution and incubated at 45 °C overnight to saturate binding, then the excess drug was removed through microcentrifugation. Spectra were measured at 45 °C on a Bruker 750 MHz spectrometer (Avance) equipped with a TCI cryoprobe. Spectra from 4 different pH conditions were processed and analyzed using NMRPipe[52] and CcpNmr Analysis[53]. Chemical shift changes for $^1$H and $^{15}$N were separately globally fit to a single p$K_a$ value using the following equation (Eq. 1):

$$\delta = \frac{\delta_H 10^{-pH} + \delta_D 10^{-pK_a}}{10^{-pH} + 10^{-pK_a}} \tag{1}$$

where $\delta_H$ and $\delta_D$ are the chemical shifts of the protonated and deprotonated states of each residue, respectively. Six residues in close proximity to E14 with large chemical shift changes with pH were analyzed. For a single p$K_a$, the modified Henserson–Hasselbach equation describes the chemical shift for each residue ($\delta$) as a function of pH[15,54,55].

**Reconstitution and preparation of solid-state NMR samples**. CDN-labeled S64V-EmrE was reconstituted into $d_{54}$-DMPC (Avanti Polar Lipids) liposomes at a protein monomer to lipid molar ratio (P: L) of 1: 25. DMPC was resuspended in 50 mM MES, 20 mM NaCl, pH 8.0 buffer at 20 mg/mL. The lipid mixture was incubated at 45 °C for 1 h to hydrate, then bath-sonicated for 1 min before addition of 0.5% octyl-glucoside (OG) followed by 30 s bath sonication. The lipid mixture was incubated at 45 °C for an additional 15 min before mixing with purified S64V-EmrE solution. After 20 min incubation at room temperature (RT), Amberlite (Supelco) was added (3 × 30 mg Amberlite per mg total detergent) to remove the detergent. The amberlite was removed after 16–24 h by simple filtration. Liposomes were collected by ultracentrifugation (165,000 × g, 6 °C, 2 h) and resuspended in a small volume (~20 mg/mL lipid concentration) of buffer. To ensure complete detergent removal, the sample was dialyzed against 1 L of the same buffer (50 mM MES, 20 mM NaCl, pH 8.0) with buffer change every 24 h over a 72 h period. The sample was then incubated with excess solid F$_4$-TPP$^+$ at RT with end-over-end rocking for at least 16 h. Excess F$_4$-TPP$^+$ was removed using microcentrifugation (7,500 × g, 5 min). Proteoliposomes were then pelleted at 100,000 × g, 4 °C, 2 h in an ultracentrifuge. Proteoliposomes were dried to ~40% hydration by mass in a desiccator. Samples were centrifuged into 1.9 and 1.3 mm MAS rotors. Two rotors were packed: a 1.9 mm sample of CDN-EmrE sample containing ~3.5 mg protein

in ~15 mg proteoliposomes, and a 1.3 mm sample of CDN-EmrE containing ~0.9 mg protein in ~3.6 mg proteoliposomes.

**Solid-state NMR experiments**. All MAS SSNMR experiments were conducted on a 600 MHz Bruker AVANCE II spectrometer. Chemical shift assignment and protein hydration experiments were carried out under 55 kHz MAS on a 1.3 mm HCN probe, whereas 2D $^1$H–$^{15}$N resolved $^1$H–$^{19}$F REDOR experiments and 2D $^{19}$F–$^{19}$F exchange experiments were conducted under 38 kHz MAS on a 1.9 mm HFX probe. Sample temperature was controlled by matching the chemical shift of water in the proteoliposomes between samples and probes. The effective sample temperatures were estimated from the water $^1$H chemical shifts using the equation $T_{eff}$ (K) = 96.9(7.83 − $\delta_{H_2O}$) where $\delta_{H_2O}$ is the observed water chemical shift[56]. By keeping the water $^1$H chemical shift at 4.89 ppm using appropriate bearing temperatures, we maintained a constant sample temperature of 285 K for all experiments. Detailed experimental conditions are provided in Supplementary Table 6.

Eight $^1$H-detected 3D MAS correlation experiments were used to assign the chemical shifts of pH 8.0 EmrE. Supplementary Fig. 11 shows the pulse sequences of those experiments that were not included in our low-pH EmrE study[39] and the pulse sequences of the 2D water-edited experiments. The hCANH, hCOcaNH, and hcaCBcaNH experiments allow intra-residue CA, CO, and CB assignment, while the hCAcoNH, hCONH, and hcaCBcacoNH experiments allow sequential residue assignment. In addition, amide-to-amide 3D correlation was achieved using the HncacoNH (Supplementary Fig. 11c) and hNcacoNH (Supplementary Fig. 11d) experiments, which allow inter-residue H$_{i-1}$–N$_i$H$_i$ and N$_{i-1}$–N$_i$H$_i$ assignment, respectively, to further disambiguate the backbone walk[57]. The coherence transfer steps and resonance assignment connectivities of these eight $^1$H-detected experiments are summarized in Supplementary Table 1. $^{15}$N–$^{13}$C correlation experiments used specific CP for polarization transfer[58]. Under 55 kHz MAS, the double-quantum (DQ) matching condition of 30 kHz $^{13}$C and 25 kHz $^{15}$N, or 35 kHz $^{15}$N and 20 kHz $^{13}$C, was used to achieve selective $^{15}$N–$^{13}$C polarization transfer. $^{13}$CA–$^{13}$CO polarization transfer was achieved using the DQ DREAM sequence under 55 kHz MAS. Due to off-resonance effects, a short 1.3–1.6 μs trim pulse (marked as a θ pulse) was used after the spin lock to rotate the magnetization to the *XY* plane[59]. For the two CB-NH correlation experiments (Fig. S11a, b), out-and-back CA-CB-CA INEPT transfer was used, using the Q3 shaped pulse for 180° pulses to invert aliphatic coherences while not inverting CO. A 6.5 ms INEPT mixing period was used for both the creation and reconversion of the antiphase magnetization, for a total of 13 ms of transfer time. Low-power $^1$H decoupling was employed for all SSNMR experiments with either CW irradiation or the WALTZ-16 scheme[60]. These $^1$H-detected MAS NMR experiments employed MISSISSPPI to suppress the water signal, using 150–200 ms of 15 kHz irradiation[61].

Water-edited NMR experiments were carried out with 2D hNH detection by inserting a water-selective echo and water-to-protein $^1$H spin diffusion mixing period following proton excitation and before CP to $^{15}$N (Supplementary Fig. 11e). A 3.5 ms Gaussian pulse with 5% truncation and 400 points for the shape was used to selectively refocus water coherence within a 3.6 ms echo, during which all protein coherence is destroyed by T$_2$ relaxation. Due to fast MAS suppressing spin diffusion, long mixing times of 325 ms for the equilibrated $S_0$ spectrum and 30 ms for the edited $S$ spectra were needed, between which significant T$_1$ relaxation occurs. To account for relaxation between the edited and equilibrated spectra, site-resolved T$_1$ measurements were carried out through saturation recovery using the pulse sequence in Supplementary Fig. 11f. After the pre-scan delay d1, we inserted a MISSISSIPPI dephasing block to saturate all $^1$H magnetization in the sample. Following this saturation, a variable delay allows T$_1$ relaxation to occur before $^1$H excitation. The experiments were run in a constant-time fashion, where the combined pre-scan delay and $\tau_{relax}$ were set to 3, 4, or 5 s depending on the relaxation time $\tau_{relax}$ used. $^1$H–$^{19}$F REDOR measurements and 2D $^{19}$F–$^{19}$F exchange experiments were run as described previously[39] to allow for direct comparison of the data between the two samples of distinct pH.

$^{19}$F chemical shifts were externally referenced to the −122.1 ppm signal of 5F-tryptophan on the CF$_3$Cl scale[62]. $^{13}$C, and $^{15}$N chemical shifts were internally referenced to the DSS-referenced chemical shifts of the solution-state values. The G90A site was previously shown to have little chemical shift perturbation with pH[15], and was therefore chosen as the referencing site; the $^1$H and $^{15}$N values were set to 8.5 ppm and 107.4 ppm, respectively, and the $^{13}$C reference was set from the hCANH peak for this residue at 47.6 ppm.

**Chemical shift assignment**. The SSNMR spectra were processed in the Bruker Topspin 3.5 software package with zero-filling, apodization, and Fourier-transformation (FT). Spectra acquired in blocks for signal averaging were added in the time domain prior to FT using a custom Python script that utilized the NMRGlue and NumPy Python packages[63,64]. Chemical shift assignment and 3D spectral plotting were performed in NMRFAM-Sparky[65]. 1D and 2D correlation spectra were plotted using Bruker Topspin's XWINPLOT. Comparisons of chemical shifts between protein monomers and samples were computed in Python and plotted with Matplotlib[66]. The asymmetry of protein monomer chemical shifts was

calculated using composite $H^N$ and $N^H$ chemical shifts according to Eq. 2:

$$\Delta\omega_{NH} = \sqrt{\frac{1}{2}[\Delta\omega_H^2 + (0.10 * \Delta\omega_N)^2]} \quad (2)$$

Here $\Delta\omega_{H/N}$ is the $^1H$ and $^{15}N$ chemical shift difference between monomers A and B. The composite CA and CO chemical shift difference was calculated similarly but without the factor of 0.10 for the gyromagnetic ratio difference. Protein $\phi$, $\psi$, and $\chi_1$ torsion angles were predicted using the TALOS-N software[67] based on the measured non-H chemical shifts. A deuterium isotope correction was applied to the $C\alpha$ and $C\beta$ chemical shifts.

**2D $^{19}F$–$^{19}F$ exchange analysis.** We quantified the exchange rates between four peaks in the 2D $^{19}F$–$^{19}F$ correlation spectra at both high and low pH. Peak volumes were integrated in Topspin using the same peak areas across all mixing times, and were normalized with respect to the sum of the integrated intensities of all peaks in each row. Thus, at short mixing times where most intensities reside on the diagonal, the three cross peaks should have normalized intensities near 0, while at sufficiently long mixing times, the diagonal and three cross peaks should equilibrate to similar intensities of ~0.25. In practice, due to spectral overlap, the measured intensities deviate from the ideal equilibrium values. The normalized intensities were fit to a single-exponential decay (Eq. 3) and single-exponential buildup (Eq. 4) equations for the diagonal and cross peaks, respectively:

$$I_{Diagonal}(t_{mix}|Y_0, P, k) = (Y_0 - P) * e^{-k*t_{mix}} + P \quad (3)$$

$$I_{CrossPeak}(t_{mix}|Y_0, P, k) = (P - Y_0) * (1 - e^{-k*t_{mix}}) + Y_0 \quad (4)$$

Here $Y_0$ is the initial intensity, $P$ is the plateau value, and $k$ is the exponential buildup or decay rate, and $t_{mix}$ is the mixing time. Fitting was performed in the SciPy optimization module of python[68]. Errors are $2\sigma$ for individual points, and were estimated from the signal-to-noise ratios (SNR) of the spectral peaks. The SNR for a reference peak was measured using Sparky's built-in routine using 1000 random points to calculate SNR, and the SNR for the remaining peaks was estimated from this value by scaling by the intensity ratio of the reference peak to the remaining peaks. The errors were propagated using Eq. 5[36].

$$\epsilon_i = 2 * \sigma = 2 * I_i * \sqrt{\left(\frac{1}{SNR_i}\right)^2 + \left(\frac{1}{SNR_{norm}}\right)^2} \quad (5)$$

where $I_i$ is the normalized intensity of the $i$-th peak, $SNR_i$ is the SNR of the $i$-th peak, and $SNR_{norm}$ is the relative SNR of the normalization factor for the row. The errors for the fitting parameters were determined using Monte Carlo analysis as shown before for $T_{1\rho}$ experiments[69]. Briefly, we simulated 1000 additional datasets for each decay and buildup curve by adding a random value from a Gaussian distribution centered at $\mu = 0$, $\sigma = 0.3$ (chosen to give values mostly within $\pm 1$) multiplied by the error for each point, $\epsilon_I$. In this way, we created 1000 new datasets with the points moving randomly within the error bar for each point, but with a Gaussian distribution around the actual measured value. These 1000 datasets were then fit to the same buildup or decay curves to obtain the fitting parameters for each simulated dataset. The standard deviation of the fit parameters in the Monte Carlo datasets was then used as the fitting parameter error reported.

**$^1H$–$^{19}F$ distance extraction.** 2D hNH-resolved $H^N$–$^{19}F$ REDOR distance restraints were determined as described before[36,37,39]. Briefly, we integrated peak volumes in the 2D $^1H$–$^{19}F$ REDOR-edited hNH $S_0$ and $S$ spectra to obtain the intensity ratios $S/S_0$ for all mixing times for each protein $H^N$. Using the SIMPSON software package, we simulated the two-spin REDOR dephasing curves for distances of 3.0–15.0 Å in 0.1 Å increments, including the magnitude and asymmetry parameters of the $^{19}F$ CSA, but with default orientation[70]. Finite-pulse effects were explicitly included in the SIMPSON program. RF inhomogeneity was accounted for by simulating for pulse flip angles of 180°–145° in 5° increments, weighted by a half-Gaussian function centered at 180° and a standard deviation of 15°[36,37]. The REPULSION320 scheme with 32 gamma angles was used for powder averaging[71]. The best-fit $^1H$–$^{19}F$ distance was extracted by minimizing the RMSD between the simulated and measured $S/S_0$ values. The uncertainty in the best-fit distance was set by the same RMSD threshold of 0.2 as the previous study[39]. Distances below this RMSD value were considered significant. In cases where little to no dephasing was observed, we set the distance upper uncertainty to 40 Å, the approximate longest possible distance in the dimer (Tables S2, S4). For residues whose signals overlap in the 2D hNH spectrum, the lower-limit distance uncertainty was increased.

**Analysis of water-edited spectra under fast MAS.** Protein hydration was investigated using a water-edited 2D experiment where water $^1H$ polarization was selectively excited and transferred to the protein and detected in 2D hNH spectra (Supplementary Fig. 10e). The hydration intensities were analyzed using a modified procedure from the previously reported approach[72–74] to account for the effects of fast MAS. Because fast MAS suppresses spin diffusion, it was necessary to use longer $^1H$ mixing times for both the equilibrated $S_0$ spectrum (325 ms) and the edited $S$ spectrum (30 ms). The edited spectra were signal averaged with 3.5–4.5 times as many scans as the equilibrated spectra to obtain sufficient SNR. Site-resolved hydration intensities were calculated by dividing the integrated peak

volumes of the 30 ms $S$ spectrum by the corresponding peak volumes in the 325 ms $S_0$ spectrum. However, at 325 ms mixing, $^1H$ $T_1$ relaxation is non-negligible, causing the $S/S_0$ values to be larger than 1 for some residues. To correct for this $T_1$ relaxation, we measured 2D hNH resolved $^1H$ saturation-recovery spectra (Supplementary Fig. 10) to obtain site-specific $^1H$ $R_1$ rates at both pH values. The $^1H$ $R_1$ rates were extracted from the integrated peak volumes using the same integration areas as for the water-edited spectra. The intensities of each $H^N$ site were normalized to the maximum intensity for that site. Error bars for each point were propagated from the SNR of the spectra according to Eq. 6:

$$\epsilon_i = 2 * \sigma = 2 * I_i * \sqrt{\left(\frac{1}{SNR_i}\right)^2 + \left(\frac{1}{SNR_{norm}}\right)^2} \quad (6)$$

The saturation recovery curves were fit to a single-exponential buildup function (Eq. 7) to obtain the $R_1$ values:

$$I(t_{relax}|P, R_1) = P * (1 - e^{-R_1 * t_{relax}}) \quad (7)$$

The $R_1$ uncertainty, $\sigma_{R1}$, was estimated using the same Monte Carlo method described above for the $^{19}F$ exchange analysis. With the site-specific $R_1$ rates known, the hydration $S/S_0$ values were corrected for relaxation between the two mixing times (325 and 30 ms) according to Eq. 8:

$$H\left(\frac{S}{S_0}, R_1\right) = \frac{S}{S_0 * e^{R_1(t_2-t_1)}} = \frac{S}{S_0} * e^{R_1(t_1-t_2)} \quad (8)$$

The error of the corrected $S/S_0$ value, $H$, is a result of the errors in the $S$, $S_0$, and $R_1$ values and was calculated according to Gauss' error propagation according to Eqs. 9 and 10:

$$\sigma_{S/S_0} = \frac{S}{S_0} * \sqrt{\left(\frac{1}{SNR_S}\right)^2 + \left(\frac{1}{SNR_{S_0}}\right)^2} \quad (9)$$

$$\sigma_H = \sqrt{e^{2R_1(t_1-t_2)} * \sigma_{S/S_0}^2 + \left[\frac{S}{S_0}(t_1 - t_2) e^{R_1(t_1-t_2)}\right]^2 * \sigma_{R_1}^2} \quad (10)$$

Using this method, the water-edited intensities can be compared fairly for different residues with different $R_1$ rates and for two pH conditions.

**Structure calculation of EmrE with bound $F_4$-TPP$^+$ at high pH.** The high-pH EmrE-TPP structure was calculated in a two-stage process similar to that detailed recently[39]. $F_4$-TPP$^+$ (PDB: VCJ) was first docked into the pH 5.8 protein structure (PDB: 7JK8) using $^1H$–$^{19}F$ distance constraints measured at pH 8.0. This docking orients the ligand and disambiguates the four-fold degenerate $^1H$–$^{19}F$ constraints. The protein was then subject to all-atom molecular dynamics refinement in explicitly solvated DMPC bilayers. In both stages, the two E14 residues were modeled in the deprotonated state. Docking was performed using the HADDOCK v2.4 webserver[75,76]. The "active residues" list was set to a list of 18 protein residues between the A and B subunits which are known to with the ligand from prior biochemical data[77] and our REDOR data. The new $^1H$–$^{19}F$ REDOR distances were used as "unambiguous" constraints which are always enforced, with ranges determined by the RMSD cutoff of 0.2 (Supplementary Fig. 5). At this stage, the $^{19}F$ atoms were input as four-fold ambiguous. Both ambiguous (active-residue defined) and unambiguous (distance measurement defined) constraints used default HADDOCK energy weighting values of 10.0, 10.0, 50.0, and 50.0 for the hot, cool1, cool2, and cool3 stages of the docking simulations. Docking was performed in DMSO and started with 1000 structures, from which 200 were outputted after refinement. These 200 structures were analyzed against the four-fold ambiguous $^1H$–$^{19}F$ distance constraints using an integrated Python-Pymol script[63,68] to select the structures that best agree with the experimental data (Supplementary Table 2). We scored the structures by the lowest sum-total of violations and the least number of violations to create two separate ensembles (Supplementary Fig. 6a and Supplementary Table 3). The lowest violation by each criterium was used to structurally disambiguate the $^1H$–$^{19}F$ pairs to create 387 distance constraints from the 116 dipolar coupling measurements[39] (Supplementary Table 4). The majority (92) of the measured dipolar couplings are weak and correspond to long distances that are four-fold degenerate. About 18 dipolar couplings are strong and can be assigned to unambiguous $H^N$–F pairs based on docking. Each of the two lowest violation structures was used for further refinement by MD simulations in GROMACS.

The docked high-pH EmrE-TPP complexes were aligned to the membrane normal using the OPM webserver[78] and were inserted into explicitly hydrated DMPC bilayers with the CHARMM-GUI[79,80] membrane builder tool[81,82]. The bilayer included 224 DMPC molecules, with 114 in one leaflet and 110 in the second, and was hydrated with TIP3 water molecules[83]. The ligand forcefield was parameterized from the coordinates, and the simulation was conducted in GROMACS[84] on NMRBox virtual servers[85]. The simulation was conducted at 310 K with CHARMM36 force fields[86–89] including the WYF parameter for cation-π interactions[90]. The simulation was restrained by the protein–ligand $^1H$–$^{19}F$ distances and TALOS predicted ($\phi$, $\psi$) and sidechain $\chi_1$ torsion angles (Supplementary Fig. 2b). The restoring force for time-averaged interatomic distance restraints used was the piecewise linear force described in the online GROMACS documentation, where if $\bar{r}_{ij}$ is the time averaged distance between atoms $i$ and $j$, the

force would be proportional to $\bar{r}_{ij} - r_0$ below $r_0$, zero between $r_0$ and $r_1$, proportional to $\bar{r}_{ij} - r_1$ between $r_1$ and $r_2$, and proportional to $r_2 - r_1$ above $r_2$. Simple time-averaged distance restraints using conservative weighting were used. The time averaging constant was set to 5 ps, and the distance restraint force constant was set to 1000 kJ/(mol*nm$^2$). Dihedral restraints were also implemented with an energy weighting of 1000. The simulation started with a 5000-step minimization with position restraints, followed by 1.875 ns of equilibration over which the position restraints were progressively weakened. The production stage of the simulation was subject only to experimental constraints and was carried out in 2 fs steps for 360 and 440 ns in two runs (Supplementary Fig. 6b). Structural ensembles from the two trajectories were assembled by taking structures in 5 ns increments in the equilibrated portion of the simulation (200–400 ns for run 1, 200–360 ns for run 2) for a total of 74 structures. These structures were subjected to 5000 steps of energy minimization with position and torsion angle restraints to remove improper dihedral angles introduced by fast-timescale fluctuations. These 74 structures were scored against the original four-fold ambiguous distance constraints to select 10 structures that best agreed with the experimental data according to the sum total of distance violations (Supplementary Table 5). By chance, five of the best structures came from the first run (mean violation magnitude $0.4 \pm 0.1$ Å, mean number of violations $= 4.4 \pm 0.5$), while another five came from the second run (mean violation magnitude $0.4 \pm 0.2$ Å, mean number of violations $= 4.2 \pm 0.8$). Within each run, the ensembles are well ordered, with a mean backbone RMSD of $1.6 \pm 0.3$ Å for the 5 structures of run 1, and $2.3 \pm 0.5$ Å for the 5 structures of run 2. Between the two sub-ensembles, the variability was higher, at $2.9 \pm 0.5$ Å backbone RMSD; most of the differences are localized to the loop regions and TM4 (Supplementary Fig. 6c) where the MD simulation has few constraints: the mean backbone RMSD between the two runs for the transmembrane helices 1–3 was $2.2 \pm 0.3$ Å, lower than the variability within the sub-ensemble of run 2. The water accessibility of the binding site was examined in the fully hydrated lowest-violation complexes of the bilayer-protein-ligand system after final MD energy minimization. Water molecules whose oxygen atom is within 15 Å of any ligand atom and whose oxygen $z$-coordinate also lies within the top and bottom planes of the protein TM helical bundles are selected. For the pH 8.0 complex, the top and bottom $z$-planes were set to be between the F23B and G77A Cα atoms. For the pH 5.8 complex, the top and bottom $z$-planes were set to be between the $z$-coordinates of the Y53A and T56B Cα atoms.

**Reporting summary**. Further information on research design is available in the Nature Research Reporting Summary linked to this article.

## Data availability
Solid-state NMR chemical shifts and distance restraints have been deposited in the Biological Magnetic Resonance Bank (BMRB) with ID number 30957. The structural coordinates have been deposited in the Protein Data Bank with accession code 7SFQ. Source data are provided with this paper.

## Code availability
Python code for $^1$H–$^{19}$F REDOR analysis, structurally-based H–F pair assignment, Gromacs simulations, $^{19}$F–$^{19}$F exchange analysis, and water-edited $^1$H–$^{15}$N analysis are available upon request at meihong@mit.edu.

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

## Acknowledgements

This work is supported by NIH grants GM095839 to K.A. H-W, GM088204 to M.H., and the MIT School of Science Camplan Fund to A.A.S. and M.H. The study made use of NMR spectrometers at the National Magnetic Resonance Facility at Madison, supported by NIH grant P41 GM103399 and P41 RR002301; the Center for Magnetic Resonance, supported by P41 GM132079; and the NMRbox, supported by P41 GM111135. Equipment was purchased with funds from the University of Wisconsin-Madison, NIH (S10RR02781, S10RR08438, S10RR023438, S10RR025062, S10RR029220), the NSF (DMB-8415048, OIA-9977486, BIR-9214394), and the USDA.

## Author contributions

M.H. and K.A.H.-W. designed the experiments. A.A.S. conducted the solid-state NMR experiments, data analysis, and structure calculation; P.S. purified the protein and

prepared the membrane samples; A.J.D. contributed to the solid-state NMR experiments and data analysis. All authors discussed the results and wrote the paper.

## Competing interests

The authors declare no competing interests.
