## [Peer Review File · Nature Communications]

High-pH Structure of EmrE Reveals the Mechanism of Proton-Coupled Substrate TransportREVIEWER COMMENTS

Reviewer #1 (Remarks to the Author):

Structural knowledge of ErmE is imperative to elucidate the mode of action of SMR transporters and to develop inhibitors in the context of antimicrobial resistance. What is more, EmrE serves a model for understanding proton-coupled transport.

This study by Hong and co-workers presents high-quality insights into the high pH structure of EmrE, which is an important milestone to understand the transport mechanism of SMR transporters in a physiologically relevant lipid environment. This is an impressive tour-de-force and merits publication in Nature Communication.

Especially the ^{19}F -based data open new vistas for solid-state NMR and add a distance dimension that was previously unheard of in our community. We will see ^{19}F -based bio-ssNMR applications more often in the future, and the here presented data (along with the previous study by Hong/Henzler-Wildman) are among the very best that I have seen.

I have some comments that I would like to see addressed:

1. Figure 4: There seems some confusion regarding the perspective and/or colouring of subunits A and B. Why is the blue subunit in front in c) and in the back in d)? It seems that a 270° rotation was applied to a), yet a 90° rotation to b). Please align the representation.
2. I do not have practical experience in ^{19}F - ^{19}F spin diffusion, and this applies to the vast majority of the biological ssNMR community. It would hence instil trust in the validity of the ^{19}F - ^{19}F exchange data if the authors showed the data obtained at 265 K, where no exchange was observed. This was done in the previous low pH EmrE paper, but the longitudinal exchange/transfer time was longer in this manuscript. Please also provide cross-sections at the relevant peak positions. This could be done in the Supporting Information.
3. Please provide a detailed violation analysis for the structure determinations in the Supporting Information. Please also provide more information on how the distance restraints were implemented in HADDOCK, what were upper and lower boundaries, what were the force constants. It would also be useful to provide this kind of information for the MD simulations.
4. It is intriguing to see that the dynamics of the substrate increases at high pH, although the binding affinity for TPP+ is higher at high pH. Given that drug binding specificity usually rather relates to enthalpic contributions and given that the substrate specificity of EmrE is apparently low, I would expect that entropic contributions play a considerable role in substrate binding/binding affinity. Could the authors comment on this? Are ITC data known for EmrE?

Markus Weingarth

Reviewer #2 (Remarks to the Author):

Shcherbakov and co-authors determined a dimeric SMR transporter EmrE in a complex with F4- TPP+ at high pH (two E14 sites - deprotonated). They also determined recently determined another state of EmrE, with F4- TPP at low pH (one of the E14 should be protonated). By comparison with two structures, they discuss the structural changes and proton-substrate coupling mechanism.

General comments

EmrE has been well-characterized as a model system of the SMR transport family. This paper found that EmrE has a more elongated and spacious binding pocket in the high-pH, and the substrate is more dynamic than in the low-pH. They also found that the dissociation of protons induces a small opening on either side of the membrane. They propose that these protonation/deprotonation-dependent conformational differences explain how proton plays a role in conformation changes coupling with substrate release in EmrE.

High-resolution structural data of EmrE are already available recently by the same group (Shcherbakov et al. Nat Commun. – Ref39), but this time "the missing link structure – a substrate-bound form in the high-pH condition" was determined. Therefore, the value of this paper should be more properly assessed by pairing together with a low-pH structure. Considering its importance as a potential future drug target, and a deeper insight into the transport mechanism of EmrE is of broad interest to the scientific community. The author's group is an expert on EmrE using solution or solid-state NMR technique; thus, their accumulated functional and structural analyses are sound.

Although there is still an argument of precise pKa value of E14 residues (e.g., Ref 21, 30 32), the results are expected to be largely in line with the results that have been obtained so far, such as asymmetric protonation, two independent pKa of E14...

In my view, the discussion and conclusions derived from the experiment are reasonable to present. I enjoyed reading and but I have some questions and suggestions that could improve the manuscript.

Page 5.

The authors discuss the hydration level in the binding pocket. This is shown by water-edited 2D ¹H NMR spectra. The authors are also experts in MD simulation, so it seems like the environment is ready. How about simulating how water molecules affect the conformation of the TM helices and the volume change of the binding pocket? The outcome might support further "The fact that these significant structural changes result exclusively from a pH change is remarkable. page 5"

Page 6. In the final paragraph, the author mentions

> This suggests that after a toxic substrate has bound to the protein from the cytoplasm and EmrE adopts a conformation similar to Fig. S7a, there is sufficient space for protons to enter and bind E14 from the periplasmic side. This could then trigger a conformational change to a state similar to our low-pH structure, with the transporter open to the periplasmic (low pH) side of the membrane. In this state, the toxic substrate is bound more peripherally to the central transport pore and is primed for release in accord with the enhanced off rate in the drug- and proton-bound state.

This is an exciting mechanistic proposal. Is it possible to add any supportive evidence from previous results or simulations? e.g., if you run the MD simulation using a high-pH structure as a starting model and then protonate one or both of E14 sites, does the protonation induce conformational changes and makes the substrate possible to be released to the periplasmic side?

Minor comments

Results – first paragraph

the pKa of S64V-EmrE – do you mean the pI value? Alternatively, pKa value of specific residue, please specify.

Reviewer #3 (Remarks to the Author):

In this paper, the authors present the high pH structure of the small multidrug transporter EmrE.

EmrE has been a paradigm for secondary drug-protein antiporters but it raised many questions with respect

to structure, oligomerisation, transport cycle and stoichiometry. There is now converging evidence that EmrE

is an asymmetric, antiparallel dimer with two Glu (E14) coordinating substrate and proton binding.

In a previous paper, the authors have presented the low pH structure of EmrE in lipid bilayers based on solid-state NMR. Here, they extend this work towards the high pH structure. Both structures are essential to understand the transport cycle. The experimental strategy is rather elegant and is based on two approaches: (i) High spectral resolution is achieved by the S64V mutant, which is fully functional but shifts the protein dynamics into a more favourable region. (ii) Highly quantitative distance restraints for the 3D structure determination have been obtained by using a ^{19}F -labelled ligand through which many ^{19}F - ^{13}C distances between ligand and protein have been measured.

The obtained structure agrees with available biochemical data and shows interesting differences to the low pH state. In particular, the binding pocket appears elongated, spacious and hydrated. Overall, the protein structure is more symmetric. The binding pocket appears to be open partially open to either side. It is amazing to see that a pH change causes such large differences.

Overall, I consider the experimental approach as elegant and highly appropriate, the data are of high quality and I fully agree with the conclusions drawn from the results.

Reviewer #1 (Remarks to the Author):

Structural knowledge of ErmE is imperative to elucidate the mode of action of SMR transporters and to develop inhibitors in the context of antimicrobial resistance. What is more, ErmE serves a model for understanding proton-coupled transport.

This study by Hong and co-workers presents high-quality insights into the high pH structure of ErmE, which is an important milestone to understand the transport mechanism of SMR transporters in a physiologically relevant lipid environment. This is an impressive tour-de-force and merits publication in Nature Communication.

Especially the ^{19}F -based data open new vistas for solid-state NMR and add a distance dimension that was previously unheard of in our community. We will see ^{19}F -based bio-ssNMR applications more often in the future, and the here presented data (along with the previous study by Hong/Henzler-Wildman) are among the very best that I have seen.

I have some comments that I would like to see addressed:

1. Figure 4: There seems some confusion regarding the perspective and/or colouring of subunits A and B. Why is the blue subunit in front in c) and in the back in d)? It seems that a 270° rotation was applied to a), yet a 90° rotation to b). Please align the representation.

We thank the reviewer for this question. The previous bottom view for the high-pH complex and A/B designation were correct. The confusion about the A/B designation and orientations of the dimer arises from the fact that alternating access switches the definition of subunit A and subunit B between the two pH states! In other words, the same molecule switches their assignment. To minimize this confusion, we believe it is to maintain the functional definition that the "top" side of the TM helical bundle is the acidic periplasmic side and the "bottom" side is the neutral cytoplasmic side. Therefore, we have now chosen to show the top view of both the high pH and low pH structures. For the same reason, in the side views, the subunit that is closer to the reader switches from B (blue) at high pH to A (orange) at low pH, even though it is the same molecule. So the previous views were all correct. And our slightly updated figure, showing a different view at the top left, hopefully clarifies this complexity further.

We have now revised the Figure 4 caption:

"(a) Top view seen from the periplasmic side of the high-pH structure, showing the crucial binding-site residues E14, Y40, Y60 and W63. (b) Top view of the low-pH structure for comparison. Note that conformer A (beige) in the high-pH complex has changed to conformer B (purple) in the low-pH complex. This conformational change switches the designation of the two subunits between high and low pH."

2. I do not have practical experience in ^{19}F - ^{19}F spin diffusion, and this applies to the vast majority of the biological ssNMR community. It would hence instill trust in the validity of the ^{19}F - ^{19}F exchange data if the authors showed the data obtained at 265 K, where no exchange was observed. This was done in the previous low pH EmrE paper, but the longitudinal exchange/transfer time was longer in this manuscript. Please also provide cross-sections at the relevant peak positions. This could be done in the Supporting Information.

In Figure S8 we have now added 1D ^{19}F cross sections and a control 2D spectrum measured at a low sample temperature of 258 K with 100 ms mixing. There is no significant exchange at this low temperature and long mixing time. This means that the ^{19}F - ^{19}F distances within each ligand are too long to cause spin diffusion on the timescale of these 2D exchange experiments. Because the ligand geometry is fixed and is independent of pH, one control spectrum at low temperature is sufficient to prove that all cross peaks seen at high temperature arise from dynamic interconversion at both pH. We have now clarified this point in the caption:

"(c) Two cross sections from the 2D spectra at both pH, showing the signal-to-noise ratios. (d) 100 ms 2D ^{19}F - ^{19}F exchange spectrum of the pH 5.8 complex at a sample temperature of 258 K. No exchange cross peaks are observed, indicating that when immobilized, the inter-fluorine distances within each molecule are too long to be measured by spin diffusion on this timescale. Thus, the high-temperature cross peaks result from substrate reorientation. "

3. Please provide a detailed violation analysis for the structure determinations in the Supporting Information. Please also provide more information on how the distance restraints were implemented in HADDOCK, what were upper and lower boundaries, what were the force constants. It would also be useful to provide this kind of information for the MD simulations.

We thank the reviewer for careful reading of the manuscript and for pointing out these details were missing from the paper. We have now added Tables S2 and S4 to give the list of distance constraints used during docking and MD, respectively. We have also added Tables S3 and S5 to give the detailed violation statistics and analysis, which were crucial in the structure calculation pipeline. We have also added details to the methods section about the implementation of distance constraints in the docking and MD simulations. With these details other people should be able to reproduce our structure calculation if needed.

4. It is intriguing to see that the dynamics of the substrate increases at high pH, although the binding affinity for TPP+ is higher at high pH. Given that drug binding specificity usually rather relates to enthalpic contributions and given that the substrate specificity of EmrE is apparently low, I would expect that entropic contributions play a considerable role in substrate binding/binding affinity. Could the authors comment on this? Are ITC data known for EmrE?

Markus Weingarh

We thank the reviewer for this interesting question. We do have a significant amount of ITC data regarding binding of multiple different substrates to EmrE under different conditions. ΔG , ΔH and

ΔS all vary with the individual substrate as expected. The data that we have acquired that most directly addresses the reviewer's question was acquired for TPP+ binding to WT EmrE as a function of pH. This data was previously published (Thomas *et al*, 2018 *J Biol Chem*, Ref 18). Analysis of the relative contribution of enthalpy and entropy to overall binding affinity is complicated by the fact that drug binding is coupled to proton release. Thus,

$$\Delta H_{\text{observed}} = \Delta H_{\text{bind}} + \Delta H_{\text{ionization}}$$

Where the observed enthalpy by ITC includes both a contribution from the binding event and the enthalpy of buffer ionization as protons are absorbed/released by the buffer. We previously performed TPP+ binding experiments in multiple buffers at 4 different pH values to determine the number of protons released upon drug binding. Since different buffers have different enthalpies of ionization, this can be used to determine the number of protons released upon drug binding to EmrE. However, we can reanalyze this data to instead extract the ΔH_{bind} values by extrapolating to $\Delta H_{\text{ionization}} = 0$. This is what we now report in **Table 3**.

We first note that at high pH where there is minimal proton release upon drug binding, ΔH and ΔS are nearly identical in all buffers confirming the accuracy of this method. Second, the affinity increases with increasing pH due to increasingly favorable ΔS that overcomes a less favorable ΔH . This increasingly favorable entropic contribution to binding at high pH is exactly consistent with the observed increase in ligand dynamics and hydration in the high pH structure reported here. We thank the reviewer for prompting us to re-examine the ITC data and add this important confirmation to the manuscript.

Reviewer #2 (Remarks to the Author):

Shcherbakov and co-authors determined a dimeric SMR transporter EmrE in a complex with F4-TPP+ at high pH (two E14 sites - deprotonated). They also determined recently determined another state of EmrE, with F4-TPP at low pH (one of the E14 should be protonated). By comparison with two structures, they discuss the structural changes and proton-substrate coupling mechanism.

General comments

EmrE has been well-characterized as a model system of the SMR transport family. This paper found that EmrE has a more elongated and spacious binding pocket in the high-pH, and the substrate is more dynamic than in the low-pH. They also found that the dissociation of protons induces a small opening on either side of the membrane. They propose that these protonation/deprotonation-dependent conformational differences explain how proton plays a role in conformation changes coupling with substrate release in EmrE.

High-resolution structural data of EmrE are already available recently by the same group (Shcherbakov *et al*. *Nat Commun.* – Ref39), but this time "the missing link structure – a substrate-bound form in the high-pH condition" was determined. Therefore, the value of this paper should be more properly assessed by pairing together with a low-pH structure. Considering its importance as a potential future drug target, and a deeper insight into the transport mechanism of EmrE is of broad interest to the scientific community. The author's group is an expert on EmrE using solution or solid-state NMR technique; thus, their accumulated functional and structural analyses are sound.

Although there is still an argument of precise pKa value of E14 residues (e.g., Ref 21, 30 32), the results are expected to be largely in line with the results that have been obtained so far, such as asymmetric protonation, two independent pKa of E14...

In my view, the discussion and conclusions derived from the experiment are reasonable to present. I enjoyed reading and but I have some questions and suggestions that could improve the manuscript.

Page 5.

The authors discuss the hydration level in the binding pocket. This is shown by water-edited 2D hNH spectra. The authors are also experts in MD simulation, so it seems like the environment is ready. How about simulating how water molecules affect the conformation of the TM helices and the volume change of the binding pocket? The outcome might support further "The fact that these significant structural changes result exclusively from a pH change is remarkable. page 5"

We thank the reviewer for these important suggestions regarding the structures and mechanistic conclusions. We have added panels to show explicit water molecules in the MD simulations in Figure 6. This suggestion prompted us to find and show that the MD simulations are consistent with the hydration and 2D ¹⁹F-¹⁹F exchange NMR data. The high-pH complex has a much more hydrated binding site, with more than twice as many water molecules in the binding cavity relative to the low pH state. Further, the MD simulations support the mechanistic hypothesis, as they clearly show the high-pH state has water molecules from both sides of the membrane to access the ligand, whereas the low-pH state has water molecules only accessing from one side of the membrane. This result significantly strengthened our conclusion of the transport mechanism by EmrE.

We have added the following in the results section:

" These spectral observations are borne out by the MD equilibrated structural ensemble, as the structure calculation explicitly solvated the protein-ligand complex in lipid bilayers. **Fig. 6e** shows membrane-embedded water molecules whose oxygen atoms lie within 15 Å of any of the ligand atoms. Strikingly, the ligand-binding site is much more hydrated in the high-pH complex: a total of 69 water molecules are found in the ligand-binding pocket, and these water molecules approach the ligand from both sides of the lipid bilayer. In comparison, only 23 water molecules are found in the ligand-binding pocket in the low-pH complex; moreover they approach the ligand only from one side, the putative periplasmic side, of the lipid bilayer. "

Our discussion section now includes the following sentences:

" It is interesting to note that the high-pH structure of the EmrE-TPP complex is not fully closed on either side of the membrane. This is borne out by the water-accessibility data (**Fig. 6a-d**) and the MD simulations (**Fig. 6e**), which show larger water accessibility of the ligand-binding cavity at high pH than at low pH. Moreover, water molecules approach the ligand from both sides of the membrane at high pH, but only accesses the ligand from one side of the membrane at low pH. "

Page 6. In the final paragraph, the author mentions

> This suggests that after a toxic substrate has bound to the protein from the cytoplasm and EmrE adopts a conformation similar to Fig. S7a, there is sufficient space for protons to enter and bind E14 from the periplasmic side. This could then trigger a conformational change to a state similar to our low-pH structure, with the transporter open to the periplasmic (low pH) side of the membrane. In this state, the toxic substrate is bound more peripherally to the central transport

pore and is primed for release in accord with the enhanced off rate in the drug- and proton-bound state.

This is an exciting mechanistic proposal. Is it possible to add any supportive evidence from previous results or simulations? e.g., if you run the MD simulation using a high-pH structure as a starting model and then protonate one or both of E14 sites, does the protonation induces conformational changes and makes the substate possible to be released to the periplasmic side?

This is an interesting idea to pursue in the future but is outside the scope of the current study. The goal of the current study is to provide direct experimental data of the high-pH structure to complete the description of the alternating-access motion of EmrE as bound to a substrate. For this experimental structure determination, we do not need simulations as supporting evidence. But in the future, it will be very interesting to follow up with simulations to investigate the molecular pathways and potential intermediate states that connect the two structures during alternating access.

Minor comments

Results – first paragraph

the pKa of S64V-EmrE – do you mean the pI value? Alternatively, pKa value of specific residue, please specify.

This refers to the pK_a value of E14. We have now clarified this in the results section.

Reviewer #3 (Remarks to the Author):

In this paper, the authors present the high pH structure of the small multidrug transporter EmrE. EmrE has been a paradigm for secondary drug-protein antiporters but it raised many questions with respect to structure, oligomerisation, transport cycle and stoichiometry. There is now converging evidence that EmrE is an asymmetric, antiparallel dimer with two Glue (E14) coordinating substrate and proton binding.

In a previous paper, the authors have presented the low pH structure of EmrE in lipid bilayers based on solid-state NMR. Here, they extend this work towards the high pH structure. Both structure are essential to understand the transport cycle. The experimental strategy is rather elegant and is based on two approaches: (i) High spectral resolution is achieved by the S64V mutant, which is fully functional but shifts the protein dynamics into a more favourable region. (ii) Highly quantitative distance restraints for the 3D structure determination have been obtained by using a ¹⁹F-labelled ligand through which many ¹⁹F-¹³C distances between ligand and protein have been measured.

The obtained structure agrees with available biochemical data and shows interesting differences to the low pH state. In particular, the binding pocket appears elongated, spacious and hydrated. Overall, the protein structure is more symmetric. The binding pocket appears to be open partially open to either side. It is amazing to see that a pH change causes such large differences.

Overall, I consider the experimental approach as elegant and highly appropriate, the data are of high quality and I fully agreed with the conclusions drawn from the results.

We are glad the reviewer appreciates the quality and novelty of this study and agrees with our conclusions. No change is needed.

REVIEWERS' COMMENTS

Reviewer #1 (Remarks to the Author):

All my comments have been expertly addressed. Especially the analysis of the binding energetics has strengthened this very interesting work even further. I support publication of the manuscript in its current form and congratulate the authors.

Reviewer #2 (Remarks to the Author):

Thank you for adding the MD simulation of water molecules in the substrate-binding site. It is a fascinating result. I am also glad to see that these results support the experimental evidence.

Although I have proposed another MD simulation, these simulations may require plenty of adjustments and detailed conditional studies in some cases. As the authors have stated, I accept this may be beyond the scope of this work, and do not ask further.

Overall, I consider the experimental data and the conclusions drawn from the results attractive and reasonable, significantly contributing to understanding this efflux system.

I have no further comments.